# Adaptive Hierarchical Density-Based Spatial Clustering Algorithm for Streaming Applications

**Darveen Vijayan [1],* and Izzatdin Aziz [2]**

1 Computer and Information Sciences Department, Universiti Teknologi PETRONAS, Persiaran UTP, Seri Iskandar 32610, Perak, Malaysia
2 Center for Research in Data Science (CeRDaS), Universiti Teknologi PETRONAS, Persiaran UTP, Seri Iskandar 32610, Perak, Malaysia
* Correspondence: darveen_20000224@utp.edu.my

**Abstract:** Clustering algorithms are commonly used in the mining of static data. Some examples include data mining for relationships between variables and data segmentation into components. The use of a clustering algorithm for real-time data is much less common. This is due to a variety of factors, including the algorithm's high computation cost. In other words, the algorithm may be impractical for real-time or near-real-time implementation. Furthermore, clustering algorithms necessitate the tuning of hyperparameters in order to fit the dataset. In this paper, we approach clustering moving points using our proposed Adaptive Hierarchical Density-Based Spatial Clustering of Applications with Noise (HDBSCAN) algorithm, which is an implementation of an adaptive approach to building the minimum spanning tree. We switch between the Boruvka and the Prim algorithms as a means to build the minimum spanning tree, which is one of the most expensive components of the HDBSCAN. The Adaptive HDBSCAN yields an improvement in execution time by 5.31% without depreciating the accuracy of the algorithm. The motivation for this research stems from the desire to cluster moving points on video. Cameras are used to monitor crowds and improve public safety. We can identify potential risks due to overcrowding and movements of groups of people by understanding the movements and flow of crowds. Surveillance equipment combined with deep learning algorithms can assist in addressing this issue by detecting people or objects, and the Adaptive HDBSCAN is used to cluster these items in real time to generate information about the clusters.

**Keywords:** moving points; HDBSCAN; crowd clustering; unsupervised learning

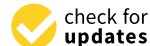



## 1. Introduction

In recent years, the number of persons accessing public services has expanded by multiple factors. This is simply due to rising urbanization, in which more people desire to live in cities and, as a result, more cities with urban amenities such as subways are being built. Because the number of people utilizing the infrastructure has risen, while the buildings and infrastructure were created much earlier, this reality has compounded the demand for better safety in public spaces. To address this issue, it has become increasingly vital to ensure that these areas are secure; some examples include subway platforms that are fully isolated from the subway lines save for the entrances. Greater monitoring of the quantity of individuals moving into the platform area can be utilized to increase the efficacy of the method [1].

Surveillance technology has become increasingly important in enhancing public safety. This is especially true in settings where people are continually congregating, such as parks, stadiums, subways, supermarkets, music halls, and airports. The sheer volume of people traveling through at all hours of the day is enormous. Movements of items can be readily tracked to guarantee safety thanks to significant developments in deep learning object

identification techniques such as Faster Region-based Convolutional Neural Network (R-CNN) and You only look once (YOLO) [2]. This, however, would necessitate staff in charge of reporting overcrowding. An algorithm that detects people's patterns and clusters would be a good addition to the present technology for improving the efficiency of the whole pipeline. Because we can simply adapt the method to numerous cameras, it would be a very scalable approach [3].

Clustering techniques are commonly employed in research to tackle data mining challenges when accurate labels are unavailable. It is one of a number of unsupervised learning approaches in which the algorithm learns patterns from a dataset. This is a good strategy for crowd movements because the actual clusters are not known ahead of time. Crowd movements are also characterized by their randomness, with no particular shape density. To ensure that the clusters are not ignored, a robust method is required. HDBSCAN is a clustering algorithm that uses an unsupervised learning algorithm. It was created by Campello, Moulavi, and Sander [4], and it extends the Density-Based Spatial Clustering Application with Noise (DBSCAN) technique by adding a hierarchical component that allows flat clusters to be extracted from the tree. The HDBSCAN method is a good unsupervised learning technique that can handle data of various densities and shapes. Because the previous distribution of the data is not known, it is critical that the method can handle data with these distributions. In addition, when running the clustering process, HDBSCAN places a greater focus on the spatial component of the points.

The rest of the paper continues as follows. Section 2 gives an overview of the HDBSCAN process' progress, followed by the first steps in constructing moving points. The Adaptive HDBSCAN technique is presented in Section 3 along with implementation details. Section 4 deals with the evaluation of the framework proposed, while Section 5 wraps up the study and suggests future research directions.

## 2. Relevant Work

In most countries, using closed-circuit television (CCTV) monitoring systems is the major method of ensuring safety in restricted areas with significant traffic [5]. Video surveillance is an inescapable part of life in today's world. It increases public safety by allowing the area to be monitored in the event of a sudden congestion. Basic systems consist of a set of cameras that send a signal to a central location where the video streams are shown on screens and recorded, and they just provide recording and display capabilities. They might either passively collect the footage or be monitored actively. CCTV is useful in a proactive approach to safety, where conditions are established to prevent accidents rather than waiting for them to happen. One technique to lessen the chances of an accident is to apply behavior-based safety (BBS) approaches [6]. Analyzing video data to uncover and highlight crucial elements for assisted scene interpretation and analysis would be a simple upgrade to the system. They recognize, categorize, track, locate, and evaluate the activity of objects of interest in the environment using computer vision algorithms or video motion detectors.

To be able to obtain insight from the movements of items in the camera frame, it is necessary to first detect such objects. There has been a lot of progress in computer vision systems that can be used to detect and extract the geolocation of points recently. These unstructured data points can be readily transferred to a data structure that machine learning algorithms can exploit to gain more insights with simple post-processing. S. Varma et al. devised a system for detecting moving objects in surveillance footage and determining whether or not they are human [7]. To account for faster object recognition, they used a well-known Background Subtraction Algorithm known as Mixture of Gaussians. To work with, the Support Vector Machine (SVM) algorithm is provided with a set of basic and efficient properties. Several SVM kernels, as well as the K Nearest Neighbor Classifier's numerous distance measurements, are used to evaluate the system's performance. The study yielded an F measure of 86.925% on average. The background subtraction algorithm is based on the

assumption that background is more often visible than the foreground, and background variance is low. This is modeled using the equation [7]

$$P(X_t) = \sum_{i=1}^{k} \omega_{i,t} \eta(X_t, \mu_{i,t}, \Sigma_{i,t}) \tag{1}$$

where

- The number of Gaussian clusters used to describe pixel history is denoted by K;
- $\omega_{i,t}$ is the weight factor associated with cluster $i$ and time $t$;
- $\mu_{i,t}$ and $\Sigma_{i,t}$ are the mean and covariance matrix of $i$-th Gaussian cluster.

With the emergence of convolutional neural networks (CNN), the capacity to bridge the gap between human and computer talents has increased dramatically. To obtain fantastic outcomes, researchers focus on a number of components in the sector. A Convolutional Neural Network (CNN) is a Deep Learning system that can take an input image, assign value to distinct objects or areas in the image using learnable weights and biases, and distinguish between them [8]. The amount of pre-processing required by a CNN is much less than that required by other classification algorithms. While early techniques include hand-engineering of filters, with adequate training, CNNs may quickly learn how to configure filters. A CNN successfully captures the Spatial and Temporal correlations in a picture by applying appropriate filters. The architecture provides better fitting to the picture dataset because of the reduced number of parameters involved and the reusability of weights. To put it another way, the network might be honed to better understand the image's complexity. A CNN, on the other hand, can only classify a single image and fails when there are many objects in the image. The length of the output layer can be changed, which is why a normal convolutional network followed by a fully connected layer will not fix the problem. That gets us to the Region-based Convolutional Neural Network (R-CNN), which is a better version of the CNN. The concept of region suggestions is central to the R-CNN series [9]. Region suggestions are used to locate items inside a photograph. To overcome the issue of selecting a high number of areas, Ross Girshick et al. proposed region recommendations, a method that uses selective search to extract only 2000 regions from a picture. As a result, instead of attempting to categorize a vast number of places, it is now possible to focus on just 2000. With further enhancements to this technology, faster R-CNN can be implemented. It works similarly to R-CNN, but instead of using a selective search strategy on the feature map to produce region proposals, it uses a separate network to anticipate area ideas. After that, a Region of Interest (RoI) pooling layer is used to categorize the image within the suggested region and forecast the bounding box offset values, and the projected region proposals are reshaped. The overall speed is significantly higher than that of R-CNN. The quicker R-CNN, on the other hand, clocked in at only 18 frames per second and had a mean average precision of 62.1. Real-time is not defined as 18 frames per second [10].

The You Only Look Once, or YOLO, algorithm is used to help with this. YOLO is based on a very different concept. This network does not take a comprehensive look at the situation. Rather, it looks at areas of the image that have a high chance of containing the item. YOLO is a method for detecting objects that differs dramatically from the previously stated region-based strategies. In YOLO [11], a single neural network predicts the bounding boxes and class probabilities for these boxes.

The image is divided into grids, with bounding boxes within each grid. This is how YOLO works. For each bounding box, the network generates a class probability and bounding box offset values. To find the item in the picture, bounding boxes with a class probability greater than a threshold value are picked and used. YOLO is thousands of times faster than traditional object identification algorithms. P. Ren et al. employed the YOLO-PC method, a unique extension of the YOLO algorithm, to count persons across frames and reported a frame rate of 40 frames per second, which is fast enough to be considered real-time given that video cameras typically shoot at 20 frames per second [12]. In subsequent

iterations of the YOLO algorithm, it has managed to improve the computation speed significantly. The YOLOv4 algorithm developed by A. Bochkovskiy et al. managed to achieve 65 frames per seconds with 43.5% average precision [13].

In the literature, the use of clustering to cluster moving points is well documented. Khaing proposed a new method for categorizing moving object trajectory data that extends the k-means algorithm. In that study, the number of clusters for the k-means approach was decided by the number of dissimilar patterns found during the selection phase. The authors further urged that the centroids be initialized based on trajectory dissimilarity, citing the impact of initial centroid choice on clustering quality and the potential for dead centroids [14]. Z. Zhang also looked at using a server to perform continuous k-means computations while watching a group of moving objects. GPS device positions were transferred to a central server. The server maintained a record of these places and regularly updated the results of registered spatial queries. The authors focused on continuous k-means monitoring of moving objects after that [15].

HDBSCAN is a powerful data exploration tool. When the data in datasets have a wide range of distributions, HDBSCAN excels. It does a good job of dealing with the edge cases of other algorithms. Clusters of various densities, cluster sizes, and cluster shapes are among the examples. Furthermore, it just requires a small number of hyperparameters, which is ideal for unsupervised clustering with unclear data. HDBSCAN was employed in another study to identify trends in user behavior in a mobile app [16].

HDBSCAN was used to locate star, galaxy, and quasi-stellar object (QSO) clusters in a multidimensional color space in one study by C. Logan and S. Fotopoulou on the application of HDBSCAN for unsupervised star, galaxy, and QSO categorization. For star, galaxy, and QSO selection, they achieved F1 ratings of 98.9, 98.9, and 93.13, respectively. According to the findings, accurate categorization with HDBSCAN necessitates meticulous attribute selection. Because of the scalability of the trained model's applicability, they strongly proposed adopting their method for existing and upcoming data-rich surveys. The data were clustered using two clustering algorithms, including HDBSCAN, after the user interactions in the apps were retrieved. The HDBSCAN algorithm discovered three separate kinds of user behavior, with feature selection methods linked to user behavior picking up 73 percent of the features [17].

The advantages of HDBSCAN can be broken down into a couple main segments. HDBSCAN inherits all the advantages of the DBSCAN algorithm. Firstly, it requires very few hyperparameters, which is very intuitive. The rest of the association is learned solely from the data. The minimum cluster size is a very intuitive parameter. The HDBSCAN does not require the minimum sample parameter that is provided in DBSCAN. Aside from that, the HDBSCAN works significantly better than the partitioning method for specific use situations, such as the one we are looking at in this study, because it does not need to be preempted for the number of clusters. It also does not necessitate that the data be distributed globally. Because it is a challenging clustering technique, the Agglomerative clustering algorithm and the k-means algorithm are both unable to remove noise from data [15]. The HDBSCAN, on the other hand, is capable of filtering out noise, which is important in our study.

In conclusion, the viability of the workflow is proven. The YOLO algorithm detects objects in the camera and then outputs the data as coordinates after some processing. The coordinates are then passed into the clustering algorithm and clusters are produced. In this study, we use the HDBSCAN algorithm, which requires only a small cluster size parameter. The minimum cluster size is determined by the area of detection, which is based on an estimate of the total area. The approach for creating the dataset and clustering the points is discussed in the following section.

## 3. Methodology

In this study, we focus on the aspects of clustering points that have already been detected by a deep learning algorithm. This includes unsupervised clustering, which is

the primary method for constructing clusters. We chose synthetic data points to stress-test the system because they are designed to replicate people's movements in public settings. Because crowd motions can be fairly chaotic, the goal is for the algorithm to be able to cluster a wide range of data point distributions.

### 3.1. Data Preparation

The dataset was created to replicate individuals moving about in a public environment. Individual movement can be separated into global and local motions, which can be simulated using a prospective method and a spatial network-based route. Individual global and local movements, as well as global and local movements of groups of people, are all included in the dataset for this study. As a result, it is critical to consider a few factors. First is the distribution of the points and the standard deviation of the distribution. The points are generated using a Gaussian distribution from the ground up. We assume the Gaussian form whenever we need to express real-valued random variables whose distribution is unknown. The probability density function f(x) can be expressed as equation [18] below:

$$f(x) = \frac{1}{\sigma\sqrt{2\pi}}e^{-\frac{1}{2}(\frac{x-\mu}{\sigma})^2} \tag{2}$$

where $\sigma$ is the standard deviation and $\mu$ is the mean of the distribution. The Central Limit Theorem (CLT), which studies the sum of many random variables, is primarily responsible for this tendency. According to CLT, as the number of terms in the summation rises, the normalized sum of a number of random variables, regardless of the distribution they came from, converges to a Gaussian distribution. The movements of individual points are also modeled after the Gaussian distribution [18]. The movement of the synthetic clusters, however, are made to move in a specific direction. Each cluster will have a different density and direction of movement. For this study, the points start out at the corners of the pathway and move across to the other side of the frame. The geolocation of the points are then extracted frame-by-frame and used by the HDBSCAN algorithm for clustering.

It becomes more difficult to produce original and different material for situations with high numbers of entities as the number of people participating grows. The nondeterministic characteristics of the system and the environment must be included in realistic modeling and simulation of complex systems. With the emphasis on assessing the system's safety in abnormal or failure settings, nondeterminism is deeply ingrained. It will quickly become too time consuming to develop or alter features for each individual one by one. If we apply a set of traits to a large number of people at once, however, we risk creating undesirable artifacts on a wider scale, such as an "army-like" look with excessively uniform or periodic distributions of people or characteristics. Since simulated data are quite perfect, we built in randomization to account for some disorder in real world datasets. For the simulation in this research, the number of points are predetermined per cluster, and at every time-slice, the points are simulated to move in a random direction with a set seed within the confounds of a distance. This will ensure that the dataset mimics the actual movements of the crowd. The overall clusters will move as instructed to move as a whole. Five total clusters are simulated with different densities, numbers of points, as well as shapes of clusters.

### 3.2. Adaptive HDBSCAN Implementation

This section outlines the process of putting the suggested Adaptive HDBSCAN algorithm into practice. The canonical HDBSCAN can be broken down into five basic steps, which are the pairwise distance calculation, Euclidean to $\lambda$-space transformation, minimum spanning tree building, cluster hierarchy building, and the condensation of the cluster hierarchies and excess of mass calculation. For Adaptive HDBSCAN, the minimum spanning tree is built adaptively based on the number of points being clustered. For smaller numbers of points, Prim's algorithm [19] is used, whereas Boruvka's algorithm [20] is used for a larger number of data points.

Figure 1 below shows the pseudocode of the Adaptive HDBSCAN algorithm. The dotted area demarcates the contribution of this paper. Canonically, HDBSCAN runs on Boruvka's algorithm regardless of the number of points. The contribution of this paper is the usage of Prim's algorithm as a method to speed up computation when the number of points is low.

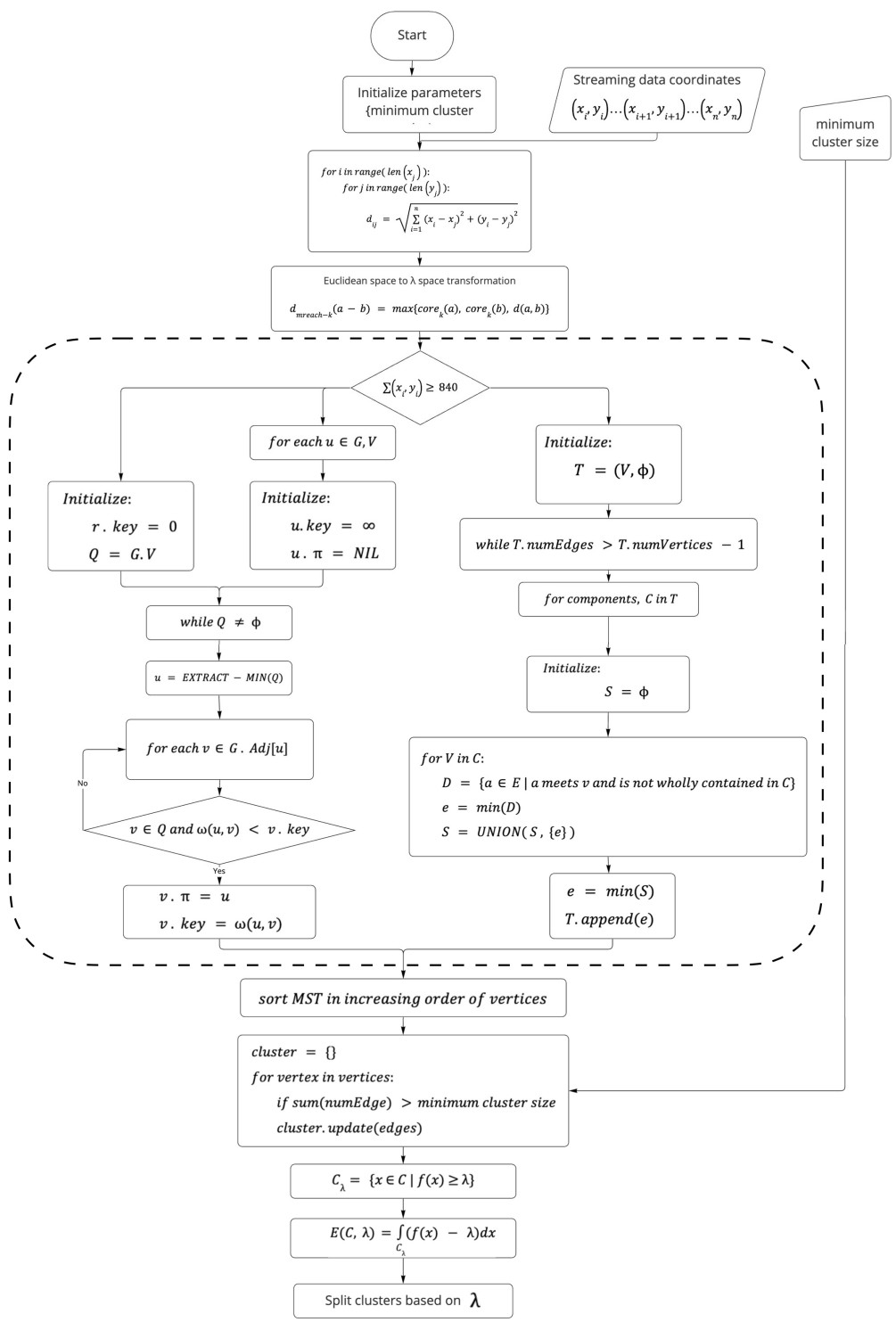

**Figure 1.** Workflow for implementing Adaptive HDBSCAN.

### 3.3. Pairwise Distance Calculation

For every slice of streaming data, the points are extracted and stored in a list. From that list, the pairwise distance between all the points are calculated. For every given pair of points, the x and y coordinates are used to find the Euclidean distance between the points. The Euclidean equation [21] used is

$$d_{i,j} = \sqrt{\sum_{i=1}^{n} (x_i - x_j)^2 + (y_i - y_j)^2} \tag{3}$$

where $d_{i,j}$ refers to the Euclidean distance between point $i$ and $j$, $(x_i, y_i)$ is the coordinate of the point $i$, and $(x_j, y_j)$ is the coordinate of the point $j$. The final product of this process is a matrix of distances between all the points in that frame. In the next process, the pairwise distance matrix is used to calculate the mutual reachability distance.

### 3.4. Euclidean to λ-Space Transformation

Using the pairwise distance matrix, the next step to clustering the points is to identify the region of points that have a higher density and find a separation between them. A good way to do this is by transforming the vector space from the Euclidean space to a density/sparsity space. In this paper, the density/sparsity space is referred to as the λ-space and the distance between the points in this new transformed space is called the mutual reachability distance. The mutual reachability distance $d_{mreach}$ is calculated using formula [21]:

$$d_{mreach-k}(a - b) = max\{core_k(a), core_k(b), d(a, b)\} \tag{4}$$

where $d(a, b)$ is the Euclidean distance between the point $a$ and $b$; $core_k(a)$ and $core_k(b)$ are the core distance for point $a$ and point $b$, respectively. Core distance for the point is defined as the radius of the smallest circle that covers $k$ points. The core distance of the points is easier understood visually, as shown in Figure 2 below.

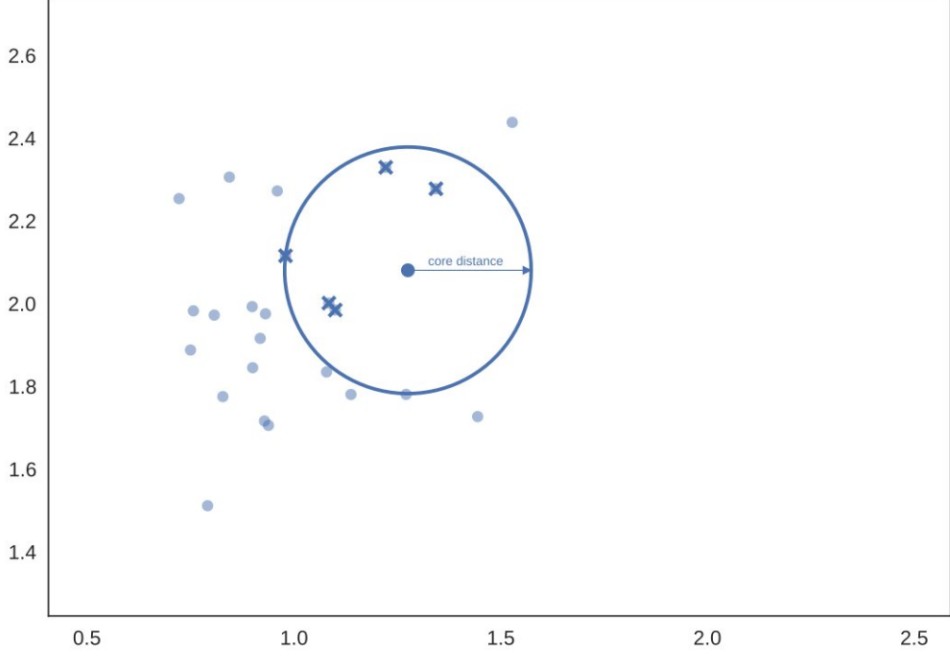

**Figure 2.** Visual representation of the core distance for the center point.

### 3.5. Determining the Cutoff for Algorithm Selection

The sparsity of the data is calculated by identifying the number of edges of the graph. For data with lower sparsity, Prim's algorithm works much better, while for data with higher sparsity, Kruskal's algorithm outperforms. Prim's algorithm runs faster in a graph with many edges as it only compares a limited number of edges per loop, whereas Kruskal's algorithm starts by sorting all the edges in the list then going through them again to check if the edge is part of the minimal spanning tree. In this paper, we approximate the sparsity of the graph using the number of points that are clustered. For more than 840 points, Kruskal's algorithm is used, whereas for the number of points below 840, Prim's algorithm is used. This is to ensure the algorithm runs optimally with minimal computational time. The number 840 is determined experimentally by a grid search to find the threshold that results in the fastest clustering computation time.

### 3.6. Prim's Algorithm

Prim's Algorithm is used to discover the shortest path between two points in a graph. Prim's approach identifies the subset of edges that includes every vertex in the graph and allows the sum of the edge weights to be reduced. Prim's algorithm starts with a single node and works its way through all of the adjacent nodes, exploring all of the connected edges along the way. There are no cycles on the edges with the smallest weights [19]. The pseudocode for Prim's algorithm is described in figure 3:

$$MST - Prim(G, \omega, r)$$

1  $for\ each\ u \in G, V :$

2  $\quad u.key\ =\ \infty$

3  $\quad u.\pi\ =\ NIL$

4  $r.key\ =\ 0$

5  $Q\ =\ G.V$

6  $while\ Q\ \neq\ \phi :$

7  $\quad u\ =\ EXTRACT - MIN(Q)$

8  $\quad for\ each\ v \in G.Adj[u] :$

9  $\quad\quad if\ v \in Q\ and\ \omega(u, v)\ <\ v.key :$

10  $\quad\quad\quad v.\pi\ =\ u$

11  $\quad\quad\quad v.key\ =\ \omega(u, v)$

**Figure 3.** Prim's algorithm pseudocode.

### 3.7. Boruvka's Algorithm

Boruvka's algorithm is the first algorithm to find the minimum spanning tree of a graph. Boruvka's algorithm was discovered by Otakar Boruvka in 1926. Boruvka's algorithm creates a spanning tree by examining each node, gradually adding edges to a developing spanning tree. Boruvka's method uses a greedy strategy, in which it finds the edge with the least weight in each iteration and adds it to the developing spanning tree [22]. The pseudocode for Boruvka's algorithm is described in figure 4:

```
MST − Boruvka(G)
  1  T = (V, ϕ)
  2  while T. numEdges > T. numVertices − 1:
  3      for components, C in T:
  4          S = ϕ
  5          for V in C:
  6              D = {a ∈ E | a meets v and is not wholly contained in C}
  7              e = min(D)
  8              S = UNION(S, {e})
  9          e = min(S)
 10          T. append(e)
```

**Figure 4.** Boruvka's algorithm pseudocode.

### 3.8. Cluster Hierarchy Construction

The minimum spanning tree is then converted into a hierarchy of connected components. In reverse order, sort the tree's edges by distance (in ascending order); then, cycle through each edge, yielding a new merged cluster. The two clusters are then compared to see if they may merge. This is accomplished through the use of a union-find data structure. A union-find data structure is also known as a disjoint-set data structure. It is a data structure for storing a group of non-overlapping (disjoint) sets. It records a partition of a set into disjoint subsets in the same way. It includes functions for creating new sets, merging sets (replacing them by their union), and locating a set's representative member. The final operation quickly determines if two components belong to the same or distinct sets. The result is a dendrogram of cluster hierarchy [21].

### 3.9. Cluster Hierarchy Condensation

In this stage, the preceding process's large and convoluted tree is condensed into a smaller tree with more data tied to each individual node. We cycle over the hierarchy, using the minimum cluster size as an important metric, to determine whether the split is just noise, or if it is greater than the minimum cluster size, we consider it a legitimate cluster split and leave it in the tree [21].

### 3.10. Excess of Mass Calculation

Finally, the excess of mass is calculated for the condensed hierarchy tree and a suitable cut is made. These are the clusters in the hierarchy with the lowest epsilon values, and they indicate clusters that cannot be divided up any more with regards to the minimum size of clusters. A new measure is developed to measure the persistence of clusters. This is denoted by $\gamma = \frac{1}{distance}$. So, for each cluster, the $\gamma_{birth}$ and the $\gamma_{death}$ are the values when the cluster splits off and becomes its own cluster. The cluster stability is then computed using equation [21]:

$$\Sigma_{p \in cluster}(\gamma_p - \gamma_{birth}) \tag{5}$$

Then, in reverse topological order, the sum of stabilities of the child clusters are computed and are compared to the stability of the cluster. If the sum of the child clusters' stabilities is larger than the cluster's stability, the cluster's stability is equal to the sum of the child clusters' stabilities. If the cluster's stability is larger than the total of its offspring, the cluster is declared a chosen cluster, and all of its descendants are unselected. We call the current collection of chosen clusters our flat clustering and return it once we reach the

root node [21]. To make this explicit, Hartigan, as well as Muller and Sawitzki, discuss the concept of mass excess where the C is a subset of a domain function f and the excess of mass of C at level is

$$E(C, \lambda) = \int_{C_\lambda} (f(x) - \lambda) dx \text{ where } C_\lambda = \{x \in C | f(x) \geq \lambda\} \tag{6}$$

So, for a cluster tree of $f$, the definition of excess of mass of a cluster $C_i$ that exists at level $\lambda_{C_i}$ of the cluster tree is $E(C, \lambda) = \int_{C_i} (f(x) - \lambda_{min}(C_t)) dx$ for $\lambda = \lambda_{min}(C_i)$ and $C_{i_1}, C_{i_2}, ... C_{i_k}$, are the children of $C_i$. Therefore, the final excess of mass equation can be expressed as $E_R(C_t) = E(C_i) - \sum_{j=1}^k E(C_{i_j})$. The $\lambda$ at which the cluster persists maximally is taken as the cutoff value for that level set. There could be multiple $\lambda$ depending on the scale of the cluster persistence. This way, clusters of different densities can be identified correctly.

## 4. Experiment Results and Discussions

In this section, the Adaptive HDBSCAN is put to the test and the results are presented and discussed. The addition of the Prim's algorithm workflow and Boruvka's algorithm is compared against the canonical single tree traversal method method, which is the Boruvka algorithm implementation.

### 4.1. Experiment Setup to Assess the Execution Time Improvement of Adaptive HDBSCAN

On the basis of execution time for data points less than 840, the Adaptive HDBSCAN was compared with the canonical HDBSCAN. Boruvka's approach is used in the canonical HDBSCAN to generate the minimum spanning tree for this number of data points. The minimum-weight edge incident to each vertex of the graph is firstly calculated; then, all of those edges are added to the forest. Subsequently, it repeats a similar process of finding the minimum-weight edge from each tree constructed so far to a different tree, and adding all of those edges to the forest. Prim's algorithm is used in the Adaptive HDBSCAN, where it works by building this tree one vertex at a time, from an arbitrary starting vertex, at each step adding the cheapest possible connection from the tree to another vertex. The experiment uses the same clusters as before, and the algorithm's execution time is recorded. To ensure uniformity, the experiment was repeated 100 times and the mean value was picked. The experiment was carried out in a python environment on a Macbook Pro with an Apple M1 processor and 8 GB of RAM.

As shown in Figure 5, the red line represents the Adaptive HDBSCAN, whereas the blue line represents the HDBSCAN. In terms of execution time, the Adaptive HDBSCAN approach proposed in this paper outperforms the regular HDBSCAN algorithm. The Prim's algorithm implementation in the Adaptive HDBSCAN for building minimum spanning trees is substantially faster than the Boruvka in this use situation. The Adaptive HDBSCAN takes 0.00485 s to complete all clustering throughout the trial, whereas the HDBSCAN takes 0.00511 s to complete all clustering.

Table 1 shows the execution time comparison for HDBSCAN and the Adaptive HDBSCAN. The Adaptive HDBSCAN approach proposed in this paper outperforms the regular HDBSCAN algorithm on average for data points below 840. The Adaptive HDBSCAN takes 0.00485 s on average to complete all clustering throughout the trial, whereas the HDBSCAN takes 0.00511 s on average to do the same.

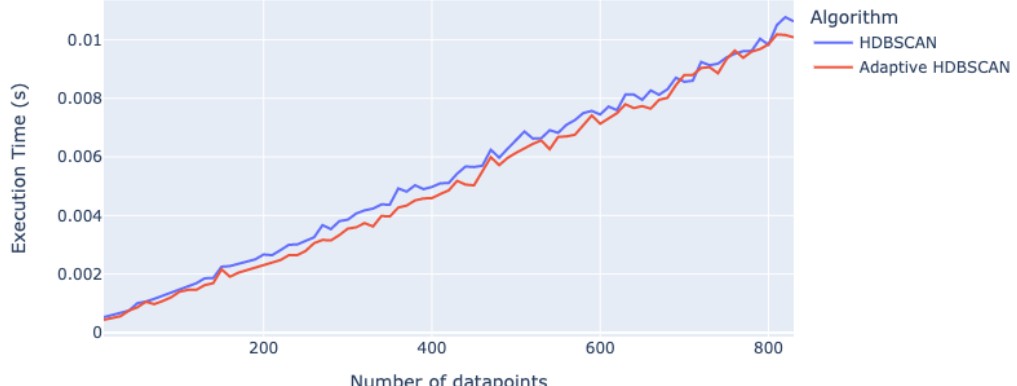

**Figure 5.** The computation time performance comparison in absolute terms between Adaptive HDBSCAN and HDBSCAN.

**Table 1.** The computation time performance comparison in absolute terms between Adaptive HDBSCAN and HDBSCAN sampled for every 100 data points increment.

| Number of Data Points | Execution Time (Seconds) | |
|---|---|---|
| | **HDBSCAN** | **Adaptive HDBSCAN** |
| 100 | 0.001490 | 0.001397 |
| 200 | 0.002667 | 0.002337 |
| 300 | 0.003855 | 0.003550 |
| 400 | 0.004973 | 0.004591 |
| 500 | 0.006595 | 0.006137 |
| 600 | 0.007446 | 0.007131 |
| 700 | 0.008563 | 0.008794 |
| 800 | 0.009830 | 0.009840 |

Figure 6 shows the percentage decrease in computation speed for the Adaptive HDB-SCAN compared with the HDBSCAN. The percentage improvement in computation speed is calculated by the following formula:

$$\frac{\text{Time taken for HDBSCAN - Time taken for Adaptive HDBSCAN}}{\text{Time taken for HDBSCAN}} \times 100\% \qquad (7)$$

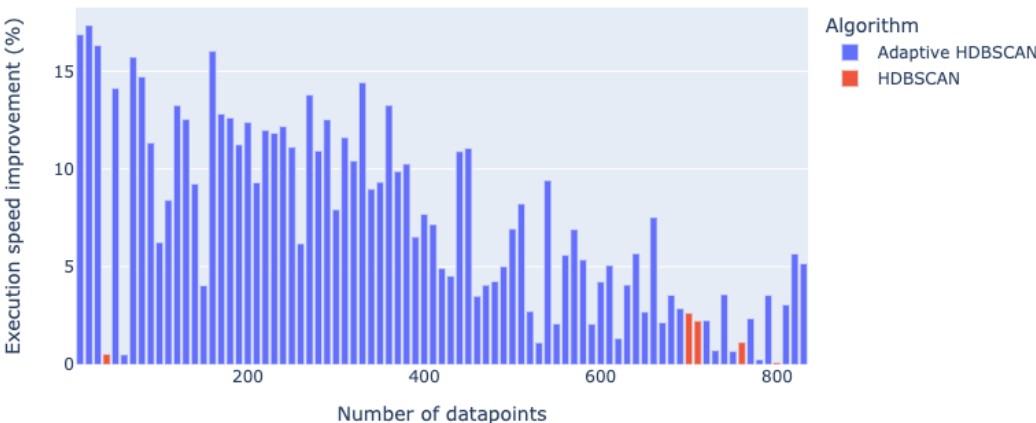

**Figure 6.** The computation time improvement of Adaptive HDBSCAN compared with HDBSCAN.

The improvements are expressed as a percentage. The maximum performance improvement is 17.5 percent, with a median improvement of 5.31 percent. Given that live streaming data continuously send out frames, this improvement is considerable. When the

amount of data points is greater than 840, the Adaptive HDBSCAN will generate the least spanning tree using Boruvka's algorithm. This is due to the fact that Boruvka's algorithm works better with a bigger number of data points. As a result, the Adaptive HDBSCAN takes advantage of all of the advantages of the original HDBSCAN while also improving the algorithm in regions where it slows down.

### 4.2. Experiment Setup to Assess the Accuracy of Clusters Generated by Adaptive HDBSCAN

The experiment to assess the accuracy of the clustering is performed by simulating a variable number of clusters and assessing the accuracy of both the Adaptive HDBSCAN and the HDBSCAN when predicting the number of clusters in the dataset. The experiment is repeated 500 times to ensure results are consistent.

Table 2 shows the table of the accuracies for both algorithms for different number of data points. The accuracy is calculated using the following formula:

$$\frac{\text{Number of accurate predictions of number of clusters per cluster set}}{500} \times 100\% \quad (8)$$

A sample of the accuracy results table is shown in Table 2. Each sample of the data point group consists of 500 experiments. The accuracy is the average across the 500 experiments. There is a slight difference between the accuracy numbers and this is caused by the excess of mass calculation, which is detailed in Section 3.10. However, in order to determine whether the difference is significant, a T-test was used to see if there is a significant difference between the means of two accuracy measurement groups.

**Table 2.** Table of accuracies of cluster prediction using the Adaptive HDBSCAN and the HDBSCAN.

| Number of Data points | Accuracy (%) | |
| :---: | :---: | :---: |
| | **Adaptive HDBSCAN** | **HDBSCAN** |
| 100 | 99.6 | 99.8 |
| 200 | 100.0 | 100.0 |
| 300 | 99.2 | 99.6 |
| 400 | 98.6 | 99.2 |
| 500 | 98.6 | 99.0 |
| 600 | 98.4 | 99.0 |
| 700 | 98.8 | 98.8 |
| 800 | 98.4 | 98.6 |

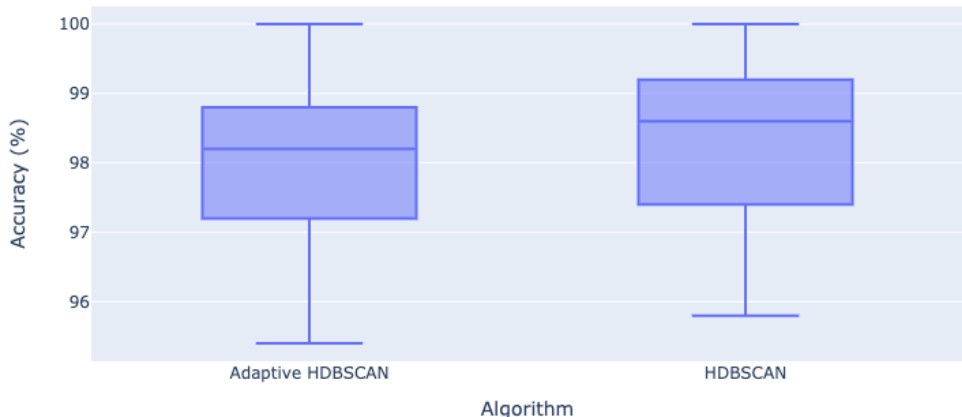

**Figure 7.** Boxplot of the accuracies of cluster prediction using the Adaptive HDBSCAN and the HDBSCAN.

A total of 100 different cluster sets are used to assess the accuracy of the prediction of the algorithm and, for each of the clusters, the experiment is run 500 times to ensure consistency. As depicted in figure 7, the Adaptive HDBSCAN has a median accuracy of 98.2%

with an upper quartile of 98.9 and a lower quartile of 97.2. The maximum and minimum accuracy are 100% and 95.4%, respectively. The HDBSCAN on the other hand has a median accuracy of 98.6% with an upper quartile of 99.2 and a lower quartile of 97.4. The maximum and minimum accuracy are 100% and 95.8%, respectively. By this measure, the HDBSCAN appears to be slightly more accurate compared with the Adaptive HDBSCAN. To further confirm if the difference was significant, a T-test is used to determine if there is a significant difference between the means of two groups of accuracy measurement. The null hypothesis is that there is no significant difference between the mean of the accuracies for both the HDBSCAN and Adaptive HDBSCAN. With a test statistic value of 1.8301 and a p-value of 0.0688, we do not have significant evidence to reject the null hypothesis since the p-value is higher than the alpha value of 0.05. Therefore, we conclude that there is no significant difference between the accuracy of the HDBSCAN and the Adaptive HDBSCAN.

Prim's algorithm performs better for less dense data points in terms of speed of execution while maintaining accuracy. This is because Prim's algorithm grows a solution from a random vertex by adding the next cheapest vertex—the vertex that is not currently in the solution but connected to it by the cheapest edge—whereas Boruvka's algorithm begins by finding the minimum-weight edge incident to each vertex of the graph, and adding all of those edges to the forest. Then, it repeats a similar process of finding the minimum-weight edge from each tree constructed so far to a different tree, and adding all of those edges to the forest.

### 5. Conclusions

The Adaptive HDBSCAN is an approach that applies a switch just before tree traversal to improve the speed of the tree traversal in data that have a smaller number of data points. This is suitable for the use case of monitoring the movement of people in a specified area. The use of Prim's algorithm is faster compared with Boruvka's algorithm when used in the HDBSCAN for a small number of data points. As shown in 3.3.1, this approach leverages all the benefits of the canonical HDBSCAN while improving the time for clusters to be detected by up to 5.3%. The Adaptive HDBSCAN takes 0.00485 s to complete all clustering, whereas the HDBSCAN takes 0.00511 s to complete all clustering. This extension will be highly useful in clustering spatial data, especially with data points that are unpredictable. In terms of accuracy, in the experiment in 3.3.2, the difference between the accuracy of the Adaptive HDBSCAN and the HDBSCAN is not significant, suggesting that the adaptive approach is viable as a solution for this use case. A p-value of 0.0688 shows that the null hypothesis, the means of the accuracy measurement from the Adaptive HDBSCAN and the HDBSCAN, are not significantly different.

**Author Contributions:** Conceptualization, D.V. and I.A; methodology, D.V.; software, D.V.; validation, D.V.; formal analysis, D.V.; investigation, D.V.; resources, D.V.; data curation, D.V.; writing—original draft preparation, D.V.; writing—review and editing, D.V.; visualization, D.V.; supervision, I.A; project administration, I.A; funding acquisition, I.A. All authors have read and agreed to the published version of the manuscript.

**Funding:** This research work is supported and funded by the Yayasan UTP grant: (015LC0-353) with title 'Predicting Missing Values in Big Upstream Oil and Gas Industrial Dataset using Enhanced 429 Evolved Bat Algorithm and Support Vector Regression', under the Center for research in Data Science (CerDaS), Universiti Teknologi PETRONAS, Malaysia.

**Institutional Review Board Statement:** Not applicable.

**Informed Consent Statement:** Not applicable.

**Data Availability Statement:** Not applicable.

**Acknowledgments:** We wish to acknowledge the tremendous support from the Department of Computer and Information Sciences (CISD), Center for Research in Data Science (CeRDaS), UTP, Malaysia for all academic support and facilities.

**Conflicts of Interest:** The authors declare no conflict of interest.

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
