# Peer review of "Adaptive Hierarchical Density-Based Spatial Clustering Algorithm for Streaming Applications"

_telecom, doi:10.3390/telecom4010001_

Round 1
Reviewer 1 Report (New Reviewer)
In this paper, adaptive hierarchical density based spatial clustering algorithm (HDBSCAN) for streaming applications is proposed and is compared with conventional hierarchical density based spatial clustering algorithm (HDBSCAN) method. This paper needs some modifications before acceptance, which are listed as follows:
1-Abstract should be rewritten with focus on the proposed work. Provide abstract with numbers and performance improvement. Just quality report not sound scientifically.
2-The first and second paragraphs of introduction should be cited with the proper references.
3-Comparison table should be added, which should contain comparison between the proposed method and other methods.
4- Quality of figures 1 and 2 should be improved, the written text in these figures is not readable.
5- References 15 to 19 not cited in correct format, which should be cited properly according to the journals template.
6- In line 125 it is mentioned that “We take a photo and divide it into grids, with bounding boxes within each grid.” This statement is about your proposed work or about reported work with YOLO algorithms? Modified this statement.
7- In Experiment Setup in section 4.1, add some explanation about Boruvka’s approach, which is used.
8- The equations which are not added by authors should be cited.
9 – It is not clearly explained that which neural network or deep learning method is presented in this paper. Also, the structure and parameters of the proposed model should be described. kindly explain this issue
10- Which data set or input data are used as streaming video? Please clarify the applied data set and also show an overview of the used data set. Also, the considered clusters should be shown and described clearly.
11- Kindly show the plot of accuracy versus number of data point for the proposed and conventional HDBSCAN methods.
12- How is it possible to obtain same accuracy for the proposed and conventional HDBSCAN methods?
Author Response
Thank you for the comments, please see the attachment for the response and updated manuscript as per recommendations

Reviewer 2 Report (New Reviewer)
The proposal is original and has the merit to be published. Nonetheless, introduction does not provide sufficient bakcground and lacks of references.
There are also few grammar errors and some references do not provide enough information.
Please refer to attached file to see all commets.

Author Response
Thank you for your comments. Please see the attachment for the response and the manuscript with changes based on recommendation.

Round 2
Reviewer 1 Report (New Reviewer)
The authors have been addressed most of my concerns. The paper can be accepted.
This manuscript is a resubmission of an earlier submission. The following is a list of the peer review reports and author responses from that submission.
Round 1
Reviewer 1 Report
The authors of the paper claim to propose adaptive dbscan for clustering. However, with current contribution, I do not see the paper is suitable for publications.
- The proposed method cannot be seen as adaptive dbscan since I do not find anything to it that is adaptive. I can only see this as an attempt to speed up the process.
- I do not see any significant improvement on that aspect as well or even it to be evaluated properly. Analysis on algorithm complexity either time or computational complexity should be conducted in addition to only recording the execution time, which may be affected by many factors when it is conducted.
- I am confused by the accuracy of the cluster here? how it is determined to be correct or not? I would suggest to use other datasets to evaluate the methods and use other clustering metrics to evaluate the methods.
- Proper comparisons with state-of-the-art methods should be conducted to appreciate the methods.
Reviewer 2 Report
1. This paper introduce some methods in Computer Vision to cluster points from streaming video data. Then make experiments on simulated data. The works are meaningful, but not very innovative, and the results are not convincing.
2. Experiments on real stream data should be added. It is not enough to test only on simulated data with a volume fo 840 points. What are the representive features for those 840 points? I cannot conduct from the experiments that this method can be applied to real camera data.
3. As said in the paper "For smaller numbers of points the prim’s algorithm is used whereas the boruvka’s algorithm is used for a larger number of data points.", how to define small or large?
4. The Figure 1. "Workflow for implementing Adaptive HDBSCAN" is hard to understand, the symbols are not explained in text.
5. The paper is verbose. Authors should put more focus onto the improvement of introduced methods, rather than their general descriptions.